# OpenReview forum: "A Survey of Recent Backdoor Attacks and Defenses in Large Language Models"
_TMLR — Accepted by TMLR_

### Review · Reviewer_ucTH · 2024-10-28

**Summary Of Contributions:**

Summary: This paper surveys the SOTA with attacks and defenses involving backdoors for large language models.

**Audience:**

Yes

**Broader Impact Concerns:**

None.

Overall: I think that this paper is useful. I think it would be reasonable to accept conditional on some expanded focus on defense and detection methods plus some simple improvements to figs/tables.

**Claims And Evidence:**

Yes

**Requested Changes:**

I would recommend some work to visually improve figs and tables. And most importantly, I would recommend expanding the scope of the paper to engage with defenses/detection methods in significantly more depth.

**Strengths And Weaknesses:**

S1: A number of surveys on backdoors have been written, but I don't see this as a knock on novelty. I can see this kind of work being valuable for its recency and focus on LLMs. I do not believe that I have seen a paper on LLM backdoors in the past. Having one of these to reference and cite would have been useful to me in the past.

W1: I think that the authors should consider citing these three papers: https://arxiv.org/abs/2401.05566, https://arxiv.org/abs/2407.15549, and https://arxiv.org/abs/2404.14461. Also consider adding to section 4 that backdoor discovery is often used as a task to study capability elicitation methods alongside jailbreaks and attacks on machine unlearning.

W2: Figure 1 is kind of messy. I don't think that the llama icons are needed. Other simplifications could happen too. Also consider replacing figure 1 with a pair of shallow trees showing the first and second columns of figures 2 and 3. Meanwhile, I now see that there are a lot of figures with similar styles, and I find most of the messy and hard to parse standalone. Consider simplifying the figs and making the captions more descriptive.

W3: Why is section 5 "Brief"? I don't have enough familiarity with the lit, but it seems to me that there's probably a good amount of work that could be done to expand this section more. And for a survey paper on backdoors in 2024, I feel as if it's a reasonable standard to want a pretty thorough discussion of both attacks and defenses.

W4: I don't see any mention of anomaly detection in sections 5 or 6. There are a handful of papers to cite related to this including https://proceedings.mlr.press/v139/hayase21a.html.

W5: I don't love the tables and their lack of captions.

---

> ### Author Response · Authors · 2024-11-01
> **Response to Reviewer ucTH**
>
> Dear Reviewer ucTH,
>
> **Thank you for your review!** We have endeavored to address all your questions and concerns below. Please let us know if there are any aspects that we need to sufficiently clarify. **If you feel that your concerns have been satisfactorily addressed, we would be grateful if you would consider revising your decision.** Please do not hesitate to reach out with any further questions. We value your feedback and welcome any additional queries.
>
> ***
>
> **Question 1:** I think that the authors should consider citing these three papers: https://arxiv.org/abs/2401.05566, https://arxiv.org/abs/2407.15549, and https://arxiv.org/abs/2404.14461. Also consider adding to section 4 that backdoor discovery is often used as a task to study capability elicitation methods alongside jailbreaks and attacks on machine unlearning.
>
> **Response 1:** Thank you for your suggestion. These three papers: [Sleeper Agents: Training Deceptive LLMs that Persist Through Safety Training](https://arxiv.org/abs/2401.05566), [Latent Adversarial Training Improves Robustness to Persistent Harmful Behaviors in LLMs](https://arxiv.org/abs/2407.15549), and [Competition Report: Finding Universal Jailbreak Backdoors in Aligned LLMs](https://arxiv.org/abs/2404.14461), are closely related to the security of large language models and are significant for exploring model security. **We have discussed and cited these papers in the Introduction**.
>
> Additionally, following your suggestion, we have also discussed the application of backdoors in jailbreaks and machine unlearning in Section 4:
>
> >Chen et al. [1] assume that malicious instructions can serve as triggers and set the rejection response as the trigger response, thereby utilizing backdoor attacks to defend against jailbreak attacks. To defend against fine-tuning-based jailbreak attacks, Wang et al. [2] leverage backdoors to enhance the security alignment of LLMs. This approach establishes a robust association between the secret prompt and secure outputs. Liu et al. [3] propose two algorithms for implementing backdoor attacks via machine unlearning. The first algorithm does not require poisoning any training samples; instead, it involves the unlearning of a small subset of contributed data. The second algorithm requires the poisoning of a few training samples, then activates the backdoor through a malicious unlearning request. Liu et al. [4] initially extract trigger patterns from the victim model, then leverage these patterns to both reverse the backdoor and induce the model to forget the backdoor through unlearning.
>
> ***
>
> **Question 2:** Figure 1 is kind of messy. I don't think that the llama icons are needed. Other simplifications could happen too. Also consider replacing figure 1 with a pair of shallow trees showing the first and second columns of figures 2 and 3. Meanwhile, I now see that there are a lot of figures with similar styles, and I find most of the messy and hard to parse standalone. Consider simplifying the figs and making the captions more descriptive.
>
> **Response 2:** The motivation for Figure 1 is to illustrate the workflow of data poisoning backdoor attacks, while Figure 2 is designed to introduce the datasets widely used in backdoor attacks across different tasks. Figure 3 aims to present backdoor attack algorithms under various fine-tuning methods, each of which serves a distinct expressive purpose.
>
> We greatly appreciate your valuable suggestion and have made our best efforts to revise Figure 1. Please refer to page 2 of the manuscript for the latest version of Figure 1.

---

> ### Author Response · Authors · 2024-11-01
> **Response to Reviewer ucTH**
>
> **Question 3:** Why is section 5 "Brief"? I don't have enough familiarity with the lit, but it seems to me that there's probably a good amount of work that could be done to expand this section more. And for a survey paper on backdoor in 2024, I feel as if it's a reasonable standard to want a pretty thorough discussion of both attacks and defenses
>
> **Response 3:** This manuscript presents a novel perspective on backdoor attacks for LLMs by focusing on fine-tuning methods. Specifically, we systematically classify backdoor attacks into three categories: full-parameter fine-tuning, parameter-efficient fine-tuning, and no fine-tuning. Therefore, the focus of this survey is on backdoor attack algorithms targeting LLMs. **We appreciate your suggestion and have included a substantial amount of the latest defense work, covering research from 2023 and 2024**. The additional work is as follows:
>
> >Sun et al. [5] propose two defensive algorithms to defend against backdoor attacks in language models. The first algorithm changes the semantics on the target side to defend against backdoor attacks, while the other is predicated on utilizing the backward probability of generating sources from given targets. The MDP algorithm [6] leverages the masking-sensitivity differences between poisoned and clean samples as distributional anchors, enabling the identification of samples under varying masking and facilitating the detection of poisoned samples. Mo et al. [7] introduce defensive demonstrations, which are derived from an uncontaminated pool through retrieval, to counteract the adverse effects of triggers. Pei et al. [8] introduce the TextGuard algorithm, designed to defend against backdoor attacks on text classification. They theoretically demonstrate that the algorithm remains effective provided the length of the backdoor trigger remains within a specified threshold. Li et al. [9] design the AttDef algorithm targeting BadNL and InSent attacks, which identifies tokens with larger attribution scores as potential triggers. Zhang et al. [10] introduce the NLPSweep algorithm to defend against character, word, sentence, homograph, and learnable textual attacks, operating independently of prior knowledge. Zhao et al. [11] utilize training loss as anchors to identify a small number of poisoned samples. Then, they calculate the similarity between poisoned samples and other samples to identify anomalous instances. Srinivasa et al. [12] propose a unified inference stage detection algorithm that is based on the latent representations of backdoored deep networks to detect poisoned samples, demonstrating robust generalization performance. Liu et al. [13] introduce the DPoE algorithm, which features a dual-model approach: a shallow model identifies backdoor shortcuts, while the main model is designed to avoid learning these shortcuts. Zhai et al. [14] propose a noise-enhanced contrastive learning algorithm to improve model robustness. The algorithm initially generates noisy training data, and then mitigates the impact of backdoors on model predictions through contrastive learning. Zhao et al. [15] introduce a backdoor mitigation approach that leverages head pruning and normalization of attention weights to eliminate the impact of backdoors on models. LMSanitator [16] achieves significantly improved convergence performance and backdoor detection accuracy by inverting predefined attack vectors. To enhance the security of graph neural networks, Sui et al. [17] identify potential triggers and filter backdoor features by predicting label transitions based on counterfactual explanations. Mu et al. [18] leverage entropy-based purification for precise detection and filtering of potential triggers in source code while preserving its semantic information. Li et al. [19] propose a two-step backdoor attack defense algorithm, where the first step involves using model preprocessing to expose the backdoor functionality, and then applying detection and removal methods to identify and eliminate the backdoor.
>
> ***
>
> **Question 4:** I don't see any mention of anomaly detection in sections 5 or 6. There are a handful of papers to cite related to this including https://proceedings.mlr.press/v139/hayase21a.html
>
> **Response 4:** Thank you for your comment. In defending against backdoor attacks, the motivation for anomaly detection is to identify poisoned samples or triggers. **Therefore, this term aligns with poisoned sample detection, which has been discussed in Section 5**. We have added an introduction to anomaly detection in that Section.
>
> In addition, the SPECTRE algorithm [17] leverages robust covariance estimation to amplify the spectral signature of poisoned samples. Although this research focuses on poisoned sample detection, it pertains to computer vision and differs from backdoor attacks targeting large language models, thus falling outside the scope of this survey. We appreciate your suggestion, cite it in our manuscript, and discuss it accordingly.

---

> ### Author Response · Authors · 2024-11-01
> **Response to Reviewer ucTH**
>
> **Question 5:** I don't love the tables and their lack of captions.
>
> **Response 5:** Thank you for your suggestion. We have revised all the tables in our manuscript and added corresponding captions.
>
> ***
>
> **References:**
>
> [1] Chen Y, Li H, Zheng Z, et al. BaThe: Defense against the Jailbreak Attack in Multimodal Large Language Models by Treating Harmful Instruction as Backdoor Trigger[J]. arXiv preprint arXiv:2408.09093, 2024.
>
> [2] Wang J, Li J, Li Y, et al. Mitigating fine-tuning jailbreak attack with backdoor enhanced alignment[J]. arXiv preprint arXiv:2402.14968, 2024.
>
> [3] Liu Z, Wang T, Huai M, et al. Backdoor attacks via machine unlearning[C]//Proceedings of the AAAI Conference on Artificial Intelligence. 2024, 38(13): 14115-14123.
>
> [4] Liu Y, Fan M, Chen C, et al. Backdoor defense with machine unlearning[C]//IEEE INFOCOM 2022-IEEE conference on computer communications. IEEE, 2022: 280-289.
>
> [5] Sun X, Li X, Meng Y, et al. Defending against backdoor attacks in natural language generation[C]//Proceedings of the AAAI Conference on Artificial Intelligence. 2023, 37(4): 5257-5265.
>
> [6] Xi Z, Du T, Li C, et al. Defending pre-trained language models as few-shot learners against backdoor attacks[J]. Advances in Neural Information Processing Systems, 2024, 36.
>
> [7] Mo W, Xu J, Liu Q, et al. Test-time backdoor mitigation for black-box large language models with defensive demonstrations[J]. arXiv preprint arXiv:2311.09763, 2023.
>
> [8] Pei H, Jia J, Guo W, et al. Textguard: Provable defense against backdoor attacks on text classification[J]. arXiv preprint arXiv:2311.11225, 2023.
>
> [9] Li J, Wu Z, Ping W, et al. Defending against Insertion-based Textual Backdoor Attacks via Attribution[C]//Findings of the Association for Computational Linguistics: ACL 2023. 2023: 8818-8833.
>
> [10] Xiang T, Ouyang F, Zhang D, et al. NLPSweep: A comprehensive defense scheme for mitigating NLP backdoor attacks[J]. Information Sciences, 2024, 661: 120176.
>
> [11] Zhao S, Tuan L A, Fu J, et al. Exploring Clean Label Backdoor Attacks and Defense in Language Models[J]. IEEE/ACM Transactions on Audio, Speech, and Language Processing, 2024.
>
> [12] Xian X, Wang G, Srinivasa J, et al. A unified detection framework for inference-stage backdoor defenses[J]. Advances in Neural Information Processing Systems, 2023, 36: 7867-7894.
>
> [13] Liu Q, Wang F, Xiao C, et al. From Shortcuts to Triggers: Backdoor Defense with Denoised PoE[C]//Proceedings of the 2024 Conference of the North American Chapter of the Association for Computational Linguistics: Human Language Technologies (Volume 1: Long Papers). 2024: 483-496.
>
> [14] Zhai S, Shen Q, Chen X, et al. Ncl: Textual backdoor defense using noise-augmented contrastive learning[C]//ICASSP 2023-2023 IEEE International Conference on Acoustics, Speech and Signal Processing (ICASSP). IEEE, 2023: 1-5.
>
> [15] Zhao X, Xu D, Yuan S. Defense against Backdoor Attack on Pre-trained Language Models via Head Pruning and Attention Normalization[C]//Forty-first International Conference on Machine Learning.
>
> [16] Wei C, Meng W, Zhang Z, et al. Lmsanitator: Defending prompt-tuning against task-agnostic backdoors[J]. arXiv preprint arXiv:2308.13904, 2023.
>
> [17] Sui, Hao, et al. DMGNN: Detecting and Mitigating Backdoor Attacks in Graph Neural Networks. arXiv preprint arXiv:2410.14105, 2024.
>
> [18] Mu, Fangwen, et al. CodePurify: Defend Backdoor Attacks on Neural Code Models via Entropy-based Purification. arXiv preprint arXiv:2410.20136, 2024.
>
> [19] Li, Yige, et al. Expose Before You Defend: Unifying and Enhancing Backdoor Defenses via Exposed Models. arXiv preprint arXiv:2410.19427, 2024.
>
> [20] Hayase J, Kong W, Somani R, et al. Spectre: Defending against backdoor attacks using robust statistics[C]//International Conference on Machine Learning. PMLR, 2021: 4129-4139.
>
> ***
>
> **In the end, we express our sincere gratitude for your detailed comments, which have been instrumental in improving our work. We greatly appreciate your insights and look forward to further discussions. If there are any additional questions or concerns, please do not hesitate to share them with us.**

---

> > ### Comment · Reviewer_ucTH · 2024-11-01
> > **I think this paper should be accepted**
> >
> > Hi, thanks for the authors for their responsiveness. I think that the paper should be accepted.
> >
> > One final note is that in addition to Hayase et al., also cite https://www.usenix.org/conference/usenixsecurity21/presentation/tang-di and other related papers if you find any.

---

> > > ### Author Response · Authors · 2024-11-02
> > > **Reply to Reviewer ucTH**
> > >
> > > Thank you for your thoughtful review and for recognizing our work. We have added citations to relevant literature and discussions in the manuscript.

---

### Review · Reviewer_JrX1 · 2024-11-06

**Summary Of Contributions:**

The authors present a survey on backdoor attacks and defenses targeting LLMs. Their study focuses on fine-tuning methods and provides a systematic categorization of these approaches. Lastly, they identify current gaps and highlight opportunities for future research.

**Audience:**

Yes

**Claims And Evidence:**

Yes

**Requested Changes:**

* Could the authors provide some arguments for the structure in the introduction or trim down the third paragraph?
* "Additionally, we believe that future research should focus more on developing backdoor attack algorithms that without fine-tuning, which could help ensure the safe deployment of LLMs." --> Fix this sentence
* The phrase "We hope our review [...]" might be a bit to informal and could be replaced with a clear statement of the paper's objectives
* In 2.1, maybe highlight the difference between pertaining and posttraining more explicitly (since it's mentioned).
* 2.2 introduces some critical properties of backdoor attacks but as far as I could see they were not referred to in the remainder of the paper to further categorize attacks. If the purpose of introducing these elements is just to give some background, I would recommend to put less emphasize on them (i.e., not a bullet list that takes a lot of space)

**Strengths And Weaknesses:**

**Strengths**
* Extensive survey on an area of research that is currently missing similar surveys
* Important aspects from the methodological approach used by different attacks, to the necessary evaluation benchmarks, to current defenses are all covered
* Categorizations provided in the paper can help to position new works in this domain. This can, in turn, make it easier to identify contributions of novel papers in comparison to existing works.
* Claims made in the introduction are all covered by sections in the paper. However, I expected a more extensive discussion of crucial issues of current works based on the abstract and introduction.
* I believe that this survey will be helpful to the research community

**Weaknesses**

* The introduction is a bit unfocused. It starts with motivation and gives a brief introduction on the topic, but then the third paragraph feels like a shortened related work section, and it was not clear to me why specific works were highlighted. Maybe the third paragraph could be considerably shortened and only provide a brief categorization of methods? Would improve the flow to the "limitations" that are discussed in the subsequent paragraph.
* Current research aims to improve the robustness of neural networks against tampering through fine-tuning (e.g., [1]). As far as I can see this form of defense is currently missing in the paper but is arguably relevant in the current open-source research landscape.


[1] Tamirisa et al., "Tamper-resistant safeguards for open-weight llms", 2024

---

> ### Author Response · Authors · 2024-11-11
> **Response to Reviewer JrX1**
>
> Dear Reviewer JrX1,
>
> **Thank you for your review!** We have endeavored to address all your questions and concerns below. Please let us know if there are any aspects that require further clarification. **If you feel that your concerns have been satisfactorily addressed, we would be grateful if you would consider revising your decision.** Please do not hesitate to reach out with any additional questions. We appreciate your input and encourage any further queries.
>
> ***
>
> **Question 1:** The introduction is a bit unfocused. It starts with motivation and gives a brief introduction on the topic, but then the third paragraph feels like a shortened related work section, and it was not clear to me why specific works were highlighted. Maybe the third paragraph could be considerably shortened and only provide a brief categorization of methods? Would improve the flow to the "limitations" that are discussed in the subsequent paragraph.
>
> **Response 1:** Thank you for your suggestions. We reorganize the third paragraph in the Introduction. This paragraph now solely contains the necessary classification of backdoor attacks and an introduction to backdoor attacks on LLMs, which we integrate into the second paragraph:
>
> >For backdoor attacks, an intuitive objective is to manipulate the model's response when a predefined trigger appears in the input samples. Attackers are required to optimize the effectiveness of their attacks while minimizing the impact on the overall performance of the model. Specifically, attackers embed malicious triggers into a subset of the training samples to induce the model to learn the association between the trigger and the target label. In model inference, when encountering the trigger, the model will consistently predict the target label, as shown in Figure 1. The activation of backdoor attacks is selective. When the input samples do not contain the trigger, the backdoor remains dormant, increasing the stealthiness of the attack and making it challenging for defense algorithms to detect. **Existing research on backdoor attack algorithms can be categorized based on the form of poisoning into data-poisoning and weight-poisoning, and additionally based on their method of modifying sample labels into poisoned-label and clean-label attacks. With the development of LLMs, a variety of backdoor attack algorithms targeting LLMs have been proposed, which include instruction poisoning and in-context learning poisoning. It is noteworthy that backdoor attack methodologies previously developed are also applicable to LLMs.**
>
> ***
>
> **Question 2:** Current research aims to improve the robustness of neural networks against tampering through fine-tuning (e.g., [1]). As far as I can see this form of defense is currently missing in the paper but is arguably relevant in the current open-source research landscape.
>
> **Response 2:** Thank you for your suggestions. Enhancing the tamper-resistant robustness of neural networks holds constructive implications for defending against backdoor attacks, therefore we include discussions of the following works in the manuscript:
>
> >Additionally, some studies attempt to construct safeguards in LLMs to enhance their security. Cao et al. [1] leverage a robust alignment checking function to defend against potential alignment-breaking attacks. This function does not require any fine-tuning of the LLM to identify adversarial queries, which potentially could defend against instruction-based backdoor attacks. Tamirisa et al. [2] design the TAR algorithm to defend against attacks, leveraging approaches from meta-learning. This algorithm can continuously safeguard the model even after thousands of steps of fine-tuning. Liu et al. [3] explore an effective assessment framework for LLM unlearning and its applications in model safeguards. Huang et al. [4] propose a perturbation-aware alignment algorithm to mitigate the security risks posed by harmful data. This algorithm adds crafted perturbations to invariant hidden embeddings, which enhances these embeddings' resistance to attacks.
>
> ***
>
> **Question 3:** Could the authors provide some arguments for the structure in the introduction or trim down the third paragraph?
>
> **Response 3:** Thank you for your valuable suggestions. We condense the third paragraph. It now contains only the essential classification of backdoor attacks and an introduction to backdoor attacks on LLMs, which we integrate into the second paragraph. For details, please refer to Response 1.

---

> ### Author Response · Authors · 2024-11-11
> **Response to Reviewer JrX1**
>
> **Question 4:** "Additionally, we believe that future research should focus more on developing backdoor attack algorithms that without fine-tuning, which could help ensure the safe deployment of LLMs." --> Fix this sentence
>
> **Response 4:** Thank you for your suggestions; we reorganize the aforementioned sentence:
>
> >Additionally, we believe that future research should focus more on developing backdoor attack algorithms that operate without fine-tuning, which could explore more mechanisms of backdoor attacks and provide new perspectives for ensuring the safe deployment of LLMs.
>
> ***
>
> **Question 5:** The phrase "We hope our review [...]" might be a bit to informal and could be replaced with a clear statement of the paper's objectives
>
> **Response 5:** Thank you for your suggestions. We restate the objective of this paper, with the revised version as follows:
>
> >Our review systematically examines backdoor attacks on LLMs, aiming to help researchers capture new trends and challenges in this field, explore security vulnerabilities in LLMs, and contribute to building a secure and reliable NLP community. Additionally, we believe that future research should focus more on developing backdoor attack algorithms that operate without fine-tuning, which could explore more mechanisms of backdoor attacks and provide new perspectives for ensuring the safe deployment of LLMs. Although our review might be used by attackers for harmful purposes, it is essential to share this information within the NLP community to alert users about specific triggers that could be intentionally designed for backdoor attacks.
>
> ***
>
> **Question 6:** In 2.1, maybe highlight the difference between pertaining and posttraining more explicitly (since it's mentioned).
>
> **Response 6:** Thank you for your comments. The motivation for Section 2.1 is to introduce the differences between LLMs and foundational language models. Therefore, we focus solely on knowledge related to LLMs. Thank you for your suggestions; we further elaborate on the differences between perturbation and post-training:
>
> >Benefiting from advanced training methods and high-quality training data, LLMs exhibit superior performance in handling downstream tasks through fine-tuning. Pre-training and fine-tuning are two critical phases in LLM development. During pre-training, LLMs acquire general language patterns from extensive of high-quality data, establishing a broad linguistic foundation. In the fine-tuning, the model is tailored to specific tasks using smaller, targeted datasets, which enhances task-specific performance. Notably, backdoor attacks frequently target the fine-tuning phase.
>
> ***
>
> **Question 7:** 2.2 introduces some critical properties of backdoor attacks but as far as I could see they were not referred to in the remainder of the paper to further categorize attacks. If the purpose of introducing these elements is just to give some background, I would recommend to put less emphasize on them (i.e., not a bullet list that takes a lot of space)
>
> **Response 7:** Thank you for your suggestions. We revise the introduction to background knowledge about backdoor attacks in Section 2.2, simplifying unnecessary character definitions:
>
> > We present the formal definition of backdoor attacks in text classification, while this definition can be extended to other tasks in natural language processing, such as question answering and knowledge reasoning. Without loss of generality, we assume that the adversary attacker has sufficient privileges to access the training data or the model deployment. Consider a standard training dataset $D_{train}$. The attacker splits the training dataset $D_{train}$ into two subsets, including a clean set $D_{train}^{clean}$ and a poisoned set $D_{train}^{poison}$. Therefore, the victim language model is trained on poisoned dataset $D_{train}^{*}$.
>
> ***
>
> **References:**
>
> [1] Cao, Bochuan, et al. "Defending against alignment-breaking attacks via robustly aligned llm." arXiv preprint arXiv:2309.14348 (2023).
>
> [2] Tamirisa, Rishub, et al. "Tamper-resistant safeguards for open-weight llms." arXiv preprint arXiv:2408.00761 (2024).
>
> [3] Liu, Sijia, et al. "Rethinking machine unlearning for large language models." arXiv preprint arXiv:2402.08787 (2024).
>
> [4] Huang, Tiansheng, Sihao Hu, and Ling Liu. "Vaccine: Perturbation-aware alignment for large language model." arXiv preprint arXiv:2402.01109 (2024).
>
> ***
>
> **In the end, we express our sincere gratitude for your detailed comments, which have been instrumental in improving our work. We greatly appreciate your insights and look forward to further discussions. If there are any additional questions or concerns, please do not hesitate to share them with us.**

---

> > ### Comment · Reviewer_JrX1 · 2024-11-11
> > **Thank you for the detailed response**
> >
> > Thank you for your detailed reply. My concerns have been adequately addressed. Overall I believe a detailed survey is notably missing in this research domain and beneficial to the community. I recommend accepting the paper.

---

### Review · Reviewer_t9z5 · 2024-11-22

**Summary Of Contributions:**

The authors provide an overview of backdoor attacks on LLMs.

**Audience:**

Yes

**Broader Impact Concerns:**

Seems covered in the ethics statement.

**Claims And Evidence:**

Yes

**Requested Changes:**

Ideally, I would like to see each section: full, param efficient, no fine-tuning be more structured ideally each with a general framework.
Improving the presentation a bit. Some sentences can be improved for clarity.

**Strengths And Weaknesses:**

Strenghts:
- Extensive list of methods
- Taxonomy based on how the model is trained on the backdoor (fully, param efficient, not at all)
- Discussing open challenges and trends

Weaknesses:
- The paper reads more like a long list of methods in each subsection than an extensive overview
- While I like the taxonomy on the number of trained params, I would hope for a more general framework within each group: mathematically defining a broad framework of crafting backdoors and e.g. fine-tune and then instantiating the framework with the different paper approaches. One example would be: Write the framework of alignment: SFT and RLHF and then show where each paper selects what attack options.
- Writing could be improved in general. One small example is the strong focus on ASR while it is probably the easiest score to understand compared to the generative ones


Questions:
- Regarding the approach of Liu et al. page 9 :  LoRA: doesn't training a malicious LoRA qualify as fine-tuning? Then it seems to me that it should be in the parameter efficient section not the "without fine-tuning"?
- Typo on page 5? "The efficiency of LLMs". Did you mean "efficacy"?

---

> ### Author Response · Authors · 2024-11-27
> **Response to Reviewer t9z5**
>
> Dear Reviewer t9z5,
>
> **Thank you for your review!** We have endeavored to address all your questions and concerns below. Please let us know if there are any aspects that we need to sufficiently clarify. **If you feel that your concerns have been satisfactorily addressed, we would be grateful if you would consider revising your decision.** Please do not hesitate to reach out with any further questions. We value your feedback and welcome any additional queries.
>
> ***
>
> **Question 1:** The paper reads more like a long list of methods in each subsection than an extensive overview
>
> **Response 1:** Thank you for your valuable comments and suggestions. **We have reorganized the structure of the paper and added detailed discussions on the connections and differences between various methods. Additionally, we have optimized the transition sentences and logical connections to ensure the coherence and readability of the content**. For example, in Section 3.1, we have added a classification of different backdoor attack works: **Leveraging LLMs; Targeted Learning Strategies; Others**. In Section 3.2, we have categorized backdoor attacks targeting PEFT into: **Prompt-tuning; Low-Rank Adaptation; Instruction Tuning; Others**. In Section 3.3, we include: **Low-Rank Adaptation; Chain-of-Thought; In-context Learning, among others**.
>
> Furthermore, we have provided a summary and trend analysis of existing work at the end of each subsection:
>
> **For Section 3.1**:
>
> Existing studies have illustrated that the security mechanisms deployed in large language models are vulnerable, which makes them particularly susceptible to exploitation through a few malicious samples. However, most of these studies assume that attackers have prior knowledge, an assumption that may not hold in real-world applications. Therefore, the following are some trends and challenges in backdoor attacks:
>
> 1. Exploring task-agnostic or black-box scenarios for backdoor attack algorithms presents more challenging conditions and represents a trend that deserves continuous scrutiny.
>
> 2. As the number of model parameters increases, the full-parameter fine-tuning strategy also introduces additional overhead to the deployment of backdoor attacks, which significantly increases the complexity of implementing such attacks.
>
> 3. Avoiding the full-parameter fine-tuning of LLMs for the deployment of backdoor attacks, which helps maintain the models' generalizability, has emerged as a prevalent trend.
>
> **For Section 3.2**:
>
> Much like a coin has two sides, although PEFT achieves impressive performance, its potential security risks require greater attention. Previous research has clearly demonstrated the effectiveness of backdoor attacks targeting PEFT methods. Below are some trends and challenges in backdoor attacks based on parameter-efficient fine-tuning algorithms:
>
> 1. Existing work primarily focuses on classification tasks; however, a new trend is exploring backdoor attacks targeting generative tasks, such as question-answering or knowledge reasoning.
>
> 2. Unlike classification tasks, backdoor attack algorithms targeting generation tasks often require malicious modification of sample labels. Although these modifications can achieve effective attack results, they may compromise the stealthiness of backdoor attack. Therefore, exploring more covert backdoor attacks in generation tasks presents a significant challenge.
>
> **For Section 3.3**:
>
> It has been proven that attackers can manipulate model responses merely through malicious instructions or poisoned demonstration examples, which severely threaten the security of LLMs. Some new challenges and trends need attention:
>
> 1. Although existing research has demonstrated the vulnerability of security measures in large language models, exploring backdoor attacks without fine-tuning in large vision-language models or multimodal decision systems is an emerging trend.
>
> 2. Backdoor attacks based on malicious instructions and poisoned demonstration examples have proven to be effective. However, their explicit triggers are easily recognized by defense algorithms. Consequently, exploring more covert triggers in backdoor attacks without fine-tuning represents a challenge that warrants sustained attention.
>
> **Thank you for your suggestions. Please refer to the manuscript for detailed modifications, which are highlighted in blue**.

---

> ### Author Response · Authors · 2024-11-27
> **Response to Reviewer t9z5**
>
> **Question 2:** While I like the taxonomy on the number of trained params, I would hope for a more general framework within each group: mathematically defining a broad framework of crafting backdoors and e.g. fine-tune and then instantiating the framework with the different paper approaches. One example would be: Write the framework of alignment: SFT and RLHF and then show where each paper selects what attack options.
>
> **Response 2:** Thank you for your valuable comments and suggestions. **We have included the mathematical definitions related to backdoor in Section 2.3**:
>
> This section formalizes the deployment methods for backdoor attacks under different settings, which include full-parameter fine-tuning, parameter-efficient fine-tuning, and no fine-tuning. In NLP, full-parameter fine-tuning generally refers to adjusting all parameters of the pre-trained LLMs to adapt to a new task or dataset. In the context of backdoor attacks, the model is specifically updated to adapt all parameters to the poisoned dataset, as illustrated in Equation 2. As the number of model parameters increases, full-parameter fine-tuning of LLMs requires the consumption of substantial computational resources. In contrast, parameter-efficient fine-tuning (PEFT) updates only a small number of model parameters, effectively enhancing the efficiency of fine-tuning:
>
> $$\phi_p = \arg\min_{\phi} \mathbb{E}_{D^{*}} (L(f(x; \theta, \phi), y) + L(f(x'; \theta, \phi), y')),$$
>
> where $\theta$ represents the original parameters of the LLMs; $\phi$ represents the parameters of the adapter layers, which are updated during the fine-tuning. Prevalent algorithms for PEFT include LoRA, prompt-tuning, and P-tuning, among others. For instance, considering LoRA, which introduces two updatable low-rank matrices $A$ and $B$, instead of updating the LLM parameters:
>
> $$W' = W + AB,$$
>
> where $W$ represents the weight matrix of the LLM, which is frozen; $A$ is a parameter matrix of dimension $d \times r$, and $B$ is a parameter matrix of dimension $r \times d$; $AB$ stands for a low-rank matrix with rank $r$, which is significantly smaller than the rank of $W$. Thus, $\phi_p \ll \theta$, significantly reducing the consumption of computational resources.
>
> For the no fine-tuning backdoor attack algorithm, which differs from the other two fine-tuning methods, this paradigm solely leverages the intrinsic reasoning capabilities of LLMs to implement the backdoor attack:
>
> $$y' = \text{Evaluate}_{LLM}(x^{'}; \theta),$$
>
> where $x^{'}$ is the input sample containing malicious instructions or prompts, and $y'$ represents the target label. For example, in in-context learning:
>
> $$x^{query} =  (I,s(x_1,l(y_1)),...,s(x_k,l(y_k)),x),$$
>
> $$y = \text{Evaluate}_{LLM} (x^{query}; \theta),$$
>
> where $I$ represents an optional instruction, $s$ denotes the demonstration examples, and $l$ represents a prompt format function.
>
> We have included the formulas corresponding to each fine-tuning algorithm in Figure 3. **Additionally, we have supplemented this with a framework based on the Learning Paradigm, which is shown in Table 1 of the manuscript. Please refer to the manuscript for detailed modifications, which are highlighted in blue. Thank you again for your suggestions**.

---

> ### Author Response · Authors · 2024-11-27
> **Response to Reviewer t9z5**
>
> **Question 3:** Writing could be improved in general. One small example is the strong focus on ASR while it is probably the easiest score to understand compared to the generative ones
>
> **Response 3:** Thank you for your comment. **We have made comprehensive revisions to the manuscript, including improving the clarity and coherence of the paper, ensuring that each technical point is expressed more accurately and is easier to understand**.
>
> In existing backdoor attack research, ASR is the main evaluation metric for algorithm effectiveness. Therefore, we have focused on explaining the meaning of ASR and the calculation process in the manuscript.
>
> Simultaneously, your point is correct. Although backdoor attacks require attention to the ASR, to ensure the stealthiness of the attacks, it is also necessary to focus on the quality of the poisoned samples. Common evaluation metrics include Perplexity (PPL), which leverage the language model GPT-2 to calculate the perplexity of poisoned samples:
>
> $$H(p, q) = -\sum_{x \in X} p(x) \log q(x)$$
>
> $$PPL = e^{H(p, q)}$$
>
> where $p(x)$ represents the true distribution of the token $x$ in the samples, and $q(x)$ is the probability distribution of the token $x$ as predicted by the GPT-2 model. For the more detailed description, please refer to page 5 of the manuscript.
>
> ***
>
> **Question 4:** Regarding the approach of Liu et al. page 9 : LoRA: doesn't training a malicious LoRA qualify as fine-tuning? Then it seems to me that it should be in the parameter efficient section not the "without fine-tuning"?
>
> **Response 4:** Thank you for your suggestion. We have reviewed the work and revised the relevant descriptions.
>
> ***
>
> **Question 5:** Typo on page 5? "The efficiency of LLMs". Did you mean "efficacy"?
>
> **Response 5:** Thank you for your comment. We have resolved this issue:
>
> >The efficacy of LLMs has been proven in various NLP tasks, demonstrating their ability to understand and generate text in ways that are both sophisticated and contextually relevant.
>
> ***
>
> **In the end, we express our sincere gratitude for your detailed comments, which have been instrumental in improving our work. We greatly appreciate your insights and look forward to further discussions. If there are any additional questions or concerns, please do not hesitate to share them with us.**

---

> > ### Comment · Reviewer_t9z5 · 2024-11-27
> >
> > Thanks for the extensive revisions. My concerns have been partially addressed. Also leaning towards acceptance now!

---

### Author Response · Authors · 2024-11-28
**Thanks for your review**

Dear AE and reviewers:

We would like to sincerely thank the AE and the three reviewers for their thoughtful comments and suggestions, which have greatly improved the quality of our survey. **Currently, after our discussions and revisions, all three reviewers have agreed to accept our manuscript**. Thank you once again for your help!

Regards,

Authors

---

### Decision · Action_Editor_c2mU · 2024-12-29

**Recommendation:** Accept with minor revision

**Comment:**

All reviewers noted that the paper successfully consolidates the existing work while highlighting unexplored areas. Minor issues raised during review were addressed in the revised version.

One minor comment -- Should (Liu et al., 2024a) be considered as a method under the category “W/o Fine-tuning” or not? Based on the discussion between the authors and Reviewer t9z5, it seems like the authors agreed with the reviewer’s clam, but the manuscript still says it’s a “W/o Fine-tuning” method. (See Table 1 for instance.)

**Audience:**

Yes, the survey addresses an important security challenge in LLMs that aligns with TMLR’s readership. It fills a gap in the literature by surveying the latest backdoor attack approaches and offering a discussion of open challenges, which will likely interest researchers and practitioners focused on the security aspects of LLMs.

**Claims And Evidence:**

The submission provides a comprehensive, up-to-date survey of backdoor attack and defense methods in LLMs. The reviewers found that the coverage is extensive and that the core claims hold: the authors systematically classify attacks into full-parameter fine-tuning, parameter-efficient fine-tuning, and no fine-tuning, supported by a broad range of references and clear methodological details (benchmarks, metrics, etc.).

---

> ### Author Response · Authors · 2025-01-04
> **Response to Action Editor c2mU**
>
> Dear Action Editor c2mU,
>
> Thank you for your suggestions and those of all the reviewers, which have significantly improved the quality of our manuscript.
>
> Regarding the research of Liu et al. (2024a), we have reclassified it to the "Backdoor Attack based on Parameter-Efficient Fine-Tuning" section. Additionally, we have revised Table 1 and Figure 3 in the manuscript. For more details, please refer to page 9 of the manuscript.
>
> Regards,
>
> Authors